# An In Vitro Model to Assess Early Immune Markers Following Co-Exposure of Epithelial Cells to Carbon Black (Nano)Particles in the Presence of *S. aureus*: A Role for Stressed Cells in Toxicological Testing

**DOI:** 10.3390/biomedicines12010128

**Published:** 2024-01-08

**Authors:** Scott Brown, Stephen J. Evans, Michael J. Burgum, Kirsty Meldrum, Jack Herridge, Blessing Akinbola, Llinos G. Harris, Rowena Jenkins, Shareen H. Doak, Martin J. D. Clift, Thomas S. Wilkinson

**Affiliations:** 1Microbiology and Infectious Disease, Institute of Life Science, Swansea University Medical School (SUMS), Swansea SA2 8PP, UK; 2In Vitro Toxicology Group, Institute of Life Science, Swansea University Medical School (SUMS), Swansea SA2 8PP, UKm.j.d.clift@swansea.ac.uk (M.J.D.C.)

**Keywords:** epithelial cells, A549, HaCaT, carbon black (nano)particles, cytokines, *Staphylococcus aureus*, infection, particle exposure, in vitro co-exposure models

## Abstract

The exposure of human lung and skin to carbon black (CB) is continuous due to its widespread applications. Current toxicological testing uses ‘healthy’ cellular systems; however, questions remain whether this mimics the everyday stresses that human cells are exposed to, including infection. *Staphylococcus aureus* lung and skin infections remain prevalent in society, and include pneumonia and atopic dermatitis, respectively, but current in vitro toxicological testing does not consider infection stress. Therefore, investigating the effects of CB co-exposure in ‘stressed’ infected epithelial cells in vitro may better approximate true toxicity. This work aims to study the impact of CB exposure during *Staphylococcus aureus* infection stress in A549 (lung) and HaCaT (skin) epithelial cells. Physicochemical characterisation of CB confirmed its dramatic polydispersity and potential to aggregate. CB significantly inhibited *S. aureus* growth in cell culture media. CB did not induce cytokines or antimicrobial peptides from lung and skin epithelial cells, when given alone, but did reduce HaCaT and A549 cell viability to 55% and 77%, respectively. In contrast, *S. aureus* induced a robust interleukin (IL)-8 response in both lung and skin epithelial cells. IL-6 and human beta defensin (hβD)-2 could only be detected when cells were stimulated with *S. aureus* with no decreases in cell viability. However, co-exposure to CB (100 µg/mL) and *S. aureus* resulted in significant inhibition of IL-8 (compared to *S. aureus* alone) without further reduction in cell viability. Furthermore, the same co-exposure induced significantly more hβD-2 (compared to *S. aureus* alone). This work confirms that toxicological testing in healthy versus stressed cells gives significantly different responses. This has significant implications for toxicological testing and suggests that cell stresses (including infection) should be included in current models to better represent the diversity of cell viabilities found in lung and skin within a general population. This model will have significant application when estimating CB exposure in at-risk groups, such as factory workers, the elderly, and the immunocompromised.

## 1. Introduction

Carbon black (CB) is a (nano)particle with a tendency to aggregate, creating three-dimensional branched clusters and larger agglomerates through Van der Waals forces [1]. CB can be engineered to have a wide range of practical everyday uses, including rubber tyre reinforcements, printer toner, rubber soles, dry batteries, conveyor belts, and flammable fluids [2,3,4]. It has further been widely used as a surrogate of particulate matter when assessing the human health risk to air pollution [5]. The versatility of CB is supported by the production of over 11 million metric tons in 2012, and 89% of this is used in the rubber industry [6]. Although a high proportion of the world’s population are readily exposed to CB, the associated public health risk is not fully understood. The World Health Organisation considers CB a potential carcinogen (Group 2B, which also includes titanium dioxide and nickel) [7]; however, meta-analyses of large occupational health studies suggest no associated risk with lung cancer [6]. Other studies have also suggested that CB is unlikely to be directly genotoxic, or a reproductive toxicant [8]. However, recent human [9] and animal studies [10,11] suggest that age and the effects of immunosuppression may have been previously underestimated.

Human lungs and skin are two major routes for CB exposure [12]. The local cellular mechanisms of toxicity and immunity have been studied in human lung and skin cell lines, such as A549 and HaCaT, respectively. For instance, A549 and HaCaT colony formation showed significant decreases in colony size, number, and viability following CB exposure [13]. CB also increased cytotoxicity as measured by lactate dehydrogenase (LDH) release [14], and genotoxicity as assessed by micronuclei frequency in A549 cells [15]. However, despite comparable concentration ranges (0–400 µg/mL), not all studies could detect cellular toxicity, as Horie and co-workers found no effects of CB on cell proliferation and intracellular reactive oxygen species (ROS) [16]. CB also produces a strong (pro-)inflammatory response through the release of macrophage chemotaxins from epithelial cells [17] and serum [18], as well as IL-8 production in A549 cells [14]. Indeed, CB may also inhibit the antimicrobial functions of the host defense peptide, LL-37, in killing *S. aureus*, *E. coli*, and rhinovirus [19]. However, limited work has assessed how these responses may differ under stressed conditions such as during lung and skin infection.

*Staphylococcus aureus* is a major pathogen in the lungs and skin, causing significant disease burden to health services worldwide [20]. *S. aureus* affects over 150,000 patients annually in the European Union (EU), resulting in attributable extra in-hospital costs of EUR 380 million for EU healthcare systems [21]. Data from the Global Burden of Disease (GBD) Study confirm that *S. aureus* was the leading bacterial cause of death in 135 countries and associated with the most deaths in those over 15 years old [22]. *S. aureus* has evolved numerous mechanisms of host evasion [23,24], making it a successful opportunistic pathogen in those individuals that are immunosuppressed and immunocompromised [25]. In the lungs, *S. aureus* is the causative organism in community acquired and ventilator-associated pneumonia (CAP and VAP) and is also found in a high proportion of cystic fibrosis patients [26,27,28,29,30]. In the skin, imbalances in the skin microbiome resulting in *S. aureus* outgrowth are responsible for atopic dermatitis flares [31,32]. The cellular mechanisms responsible for *S. aureus* pathogenesis in skin and lung epithelial cells are now well understood [31,33,34]. A key skin response to *S. aureus* is the interaction with toll-like receptor (TLR)-1, -2, and -6, which recognize *S. aureus* cell wall lipopeptides and peptidoglycans [33]. Interaction of *S. aureus* with TLR-2 on keratinocytes can induce the production of neutrophil chemoattractants, such as IL-8, and antimicrobial peptides, including cathelicidin LL-37 together with beta defensins [33]. A key epithelial response to *S. aureus* and its associated virulence factors in the lung are dependent on TLR-2 signalling and inflammasome activation, to generate (pro-)inflammatory mediators, such as IL-1β, IL-6, and the neutrophil chemokine IL-8 [35].

Much less has been reported on the co-exposure of CB and *S. aureus*. To date, one paper demonstrates that exposure of mouse lungs through inhalation to 10 mg/m^3^ of CB for 4 days suppressed immune defence against the phagocytosis and removal of *S. aureus* 24 h after exposure [36]. Recently, Hussey and co-workers showed that ‘black carbon’ increases the thickness of *S. aureus* biofilms, increases the minimum inhibitory concentration of antibiotics needed to kill *S. aureus*, and modifies the severity and invasiveness *of S. aureus* lung infection in vivo [37]. Critically, there is a paucity of studies and models investigating the co-exposure of (nano)particles (e.g., CB) and an infectious agent (e.g., *S. aureus*). Such co-exposure studies would aim to model scenarios: (i) of CB exposure on the background of an infection; (ii) of resilience to infection having been exposed to CB. These scenarios are particularly relevant for the elderly, healthcare workers, and occupational exposure across a variety of industries.

To address this gap in the literature, the current work aims to develop a reproducible in vitro co-exposure model using CB and *S. aureus* as the test particle and infectious agent due to their global exposure to humans. Our working hypothesis for this study states that (i) co-exposure responses can be determined in a cell culture system; and (ii) that co-exposure to particle and pathogen leads to additive responses. This goal was achieved by using HaCaT, keratinocytes, and A549 type II lung adenocarcinoma cells as the epithelial cells of choice due to their robust (pro-)inflammatory responses to a variety of stimuli and their resistance to injury [33,38,39,40]. This paper demonstrates the major differences in inflammatory responses to CB when comparing ‘healthy’ and ‘stressed’ epithelial cells.

## 2. Materials and Methods

### 2.1. Bacterial Strains and Routine Culture of S. aureus

Six clinical methicillin resistant *S. aureus* isolates obtained from human bronchoalveolar alveolar lavage fluid (BAL) were used [41,42]. In addition, two control strains—methicillin-resistant *S. aureus*, SH1000, and methicillin-sensitive *S. aureus*, Cowan 1, ([43,44,45,46,47,48] and Table 1) were used. For culture, one single colony of *S. aureus* was taken from an agar plate and inoculated into 5 mL of sterile tryptic soy broth (TSB), then grown overnight at 37 °C. Then, overnight cultures were standardised to OD_600_ = 0.1 (~1 × 10^8^ cfu/mL).

### 2.2. Preparation and Characterisation of Carbon Black (CB)

#### 2.2.1. CB Dose Preparation

CB, AROSPERSE^®^ 15 thermal black powder (Evonik Degussa GmbH, Essen, Germany #BT10506621) was weighed out at 1 mg using the OHAUS Explorer Semi-Micro Balance housed in a WAYSAFE (#GP1540) to 1 mL of the selected media (ultra-pure H_2_O, Tryptic soy broth (TSB) or 1% FBS/DMEM). This suspension was vortexed for 1 min and sonicated in a 90 W ultrasonic water bath (Thermo Fisher Scientific, Altrincham, UK #FB15046) at maximum power for approximately 30–40 min to ensure the CB was completely suspended. CB was then diluted to specific doses (including 0, 2, 4, 8, 10, 25, 50, 100 µg/mL) in water, TSB, or 1% fetal bovine serum (FBS) in Dulbecco’s Minimal Essential Medium (DMEM) supplemented with 1% L-Glutamine (2 mM).

#### 2.2.2. Zetasizer

Agglomerate medial and size distribution of CB samples was determined by dynamic light scattering (DLS) using a Malvern Zetasizer Nano ZS (Malvern Instruments Ltd., Malvern, UK). Measurements were performed in deionised water, TSB, and 1% FBS DMEM and presented as an average of 10 readings (*n* = 10), with samples briefly vortexed and incubated at 37 °C prior to measurements.

#### 2.2.3. *S. aureus* Growth with CB

Standardised suspensions of *S. aureus* SH1000 were standardised to OD_600_ = 0.2 (2X) their final concentrations in water, TSB, or 1%FBS/DMEM. Likewise, CB was also prepared at twice the concentration used (0, 4, 8, 16, 20, 50, 100, and 200 µg/mL). Preliminary experiments confirmed that CB interfered with the measurement of absorbance (OD600 nm) using the microplate reader (Appendix A), and therefore, direct colony counting was undertaken. Then, using an Eppendorf for each combination, 500 μL of *S. aureus* and 500 μL of the double concentrated CB were combined (to generate the defined concentration) and left to incubate at 37 °C on a rotator at 10 rpm for either 5 or 24 h (the logarithmic and stationary phases of *S. aureus*, SH1000, respectively). Then, 5 µL of an appropriate dilution series was plated onto blood agar plates in triplicate, and the plates were allowed to dry before incubating overnight at 37 °C. The next day, colonies were counted, and results presented as cfu/mL.

### 2.3. Transmission Electron Microscopy (TEM)

TEM was used to analyse CB particle size, shape, morphology, crystallinity, and purity. A drop of diluted material (50 μg/mL in double distilled H_2_O) was drop-cast on a copper TEM grid coated with a continuous carbon film (Agar Scientific, Stansted, UK) and left to air dry overnight, in a sterile environment. TEM analysis was undertaken with a FEI Talos F200x G2 TEM (Thermo Fisher Scientific, Altrincham, UK), operating at 200 kV and fitted with a high-angle annular dark field (HAADF) detector, a Gatan Orius SC600A CCD camera, and an Oxford Instruments 80 mm^2^ silicon drift energy-dispersive X-ray (EDX) spectrometer. Images were taken from 20 areas at magnifications between ×7000 and ×40,000 with a dwell time of 10 μs.

### 2.4. Preparation of CB and S. aureus for SEM

Bacterial and CB suspensions were prepared as previously described. *S. aureus* and CB suspensions were then bound to Thermanox disks. Briefly, pre-sterilised Nunc™ Thermanox™ Coverslips (13 mm diameter, Thermo Scientific, Paisley, UK) were taped onto a glass microscope slide and placed in a cytospin filter cartridge. Suspensions were mixed gently before 80 μL was pipetted into the cytospin cartridges prior to centrifugation at 112× *g* for 3 min in a Shandon Cytospin 3. The cartridges were then taken apart and the Thermanox™ disk removed carefully. The samples were then left to air-dry on the bench overnight before being fixed with 2.5% glutaraldehyde in 0.1 M piperazine-N,N′-bis(2-ethanesulfonic acid (PIPES), pH 7.4 for 5 min, post-fixed with 1% osmium tetroxide (OsO4; Simec Trade AG, Zofingen, Switzerland) in 0.1 M PIPES (pH 6.8) for 1 h, dehydrated through an ethanol series (50%, 70%, 96%, 100%, 5 min each step) and an ethanol: hexamethyldisilazane (HMDS) series (2:1, 1:1, 1:2, for 5 min each step), and finally in 100% HMDS for 5 min. Samples were then left to air-dry overnight, mounted on stubs, and viewed with a Hitachi S4800 high-resolution scanning electron microscope using the upper secondary electron (SE) detector with a –10 V bias to SE detection and maximise backscattered electron (BSE) detection, at an acceleration voltage of 1 kV and emission current of 10 µA. Images were taken from 10 areas at magnifications between ×4000 and ×58,000.

### 2.5. Cell Culture

The choice of cells was determined by the major route of exposure for CB, namely lung and skin. Likewise, the lung and skin form the major sites for *S. aureus* pathogenesis. Therefore, epithelial cell lines from lung and skin were used in this study. HaCaT immortalised human keratinocytes, and A549 adenocarcinoma Type II human alveolar epithelial cells were used [49,50]. Both cell lines were grown in Dulbecco’s, minimum essential medium (DMEM) supplemented with 10% fetal bovine serum (FBS), 1% penicillin (100 U/mL)/streptomycin (100 µg/mL, Thermo Fisher Scientific, Altrincham, UK, Cat number 15140122), 1% glutamine (2 mM), and incubated at 37 °C/5% CO_2_. HaCaT and A549s were sub-cultured at 90% confluence with TrypLE Express according to manufacturer’s instructions. Cell viability in all experiments was assessed by trypan blue (0.2%) exclusion using a Countess™ automated cell counter and the results expressed as % viable cells.

### 2.6. S. aureus Infection of Epithelial Cells

A 10× suspension of *S. aureus* SH1000 was prepared following overnight pre-culture (OD_600_ = 1.0, ~1 × 10^9^/mL). Replicate 24 well plates were seeded with 50,000 epithelial cells/well in a total volume of 1 mL cell culture media. The plates were then incubated at 37 °C in a 5% CO_2_ environment for 24 h. Then, the following day media was removed and replaced with 995 µL of DMEM supplemented with 1% FBS and L-glutamine (without antibiotics). Then, 5 μL (~5 × 10^6^ bacteria) of 10× *S. aureus* SH1000 were added before the plate was incubated for either 5 or 24 h at 37 °C in a 5% CO_2_ environment. The media was removed and decanted into a clean Eppendorf and centrifuged at 8064× *g* for 5 min. After centrifugation, 900 μL of the supernatant was removed without dislodging the pellet and aliquoted into a clean Eppendorf before being placed in a −80 freezer for storage until ELISA determination could occur.

### 2.7. Co-Exposure of Epithelial Cells to S. aureus and CB

Epithelial cell monolayers and *S. aureus* were prepared as previously described for infection studies with *S. aureus* SH1000 alone, except the following day media was removed and replaced with 495 µL of DMEM supplemented with 1% FBS and L-glutamine (without antibiotics). Then, 5 μL (~5 × 10^6^ bacteria) of 10× *S. aureus* SH1000 and 500 μL of CB dilutions (0–200 µg/mL) were added before the plate was incubated for either 5 or 24 h at 37 °C in a 5% CO_2_. The media was then removed and decanted into a clean Eppendorf and centrifuged at 8064× *g* for 5 min. After centrifugation, 900 μL of the supernatant was removed without dislodging the pellet and aliquoted into a clean Eppendorf before being placed in a −80 freezer for storage until ELISA determination could occur.

### 2.8. Enzyme Linked Immunosorbent Assay (ELISA)

DuoSet ELISA (R&D Systems, Abingdon, UK) for human IL-8 (Cat # DY208), IL-6 (Cat # DY206), and IL-10 (Cat # DY217B) were carried out according to the manufacturers’ instructions. Human β-Defensin-2 (hβD-2) ELISA was assayed using a TMB development kit (PeproTech, London, UK), according to the manufacturer’s instructions (Cat # 900-K172).

### 2.9. Data and Statistical Analysis

Growth and cytokine data were presented as the mean ± standard error of the mean (SEM). A minimum of three biological repeats were conducted for all analyses presented. GraphPad Prism software (Version 9.1.2) was used for statistical analysis using a two-way ANOVA parametric test, including a Tukey’s post hoc test for multiple pairwise comparisons with * *p* ≤ 0.05 being considered significant. The Kruskal–Wallis and post hoc test was used where data were found to be non-parametric.

## 3. Results

### 3.1. Characterisation of Carbon Black (CB)

Firstly, the physicochemical characteristics of the CB used were determined. SEM (Figure 1) and TEM (Figure 2) confirmed both the heterogeneous nature of the particles’ morphology and the approximate size in the nanometre range. EDX spectroscopy (Appendix A) of CB detected a strong ‘carbon’ signal with minor contaminating peaks of silicon, sodium, and chlorine, providing context as to the purity of the sample.

Next, CB size characteristics were assessed in biological media supporting bacterial and epithelial cell growth. Considering that bacterial and cell culture growth media contain significant additional protein sources to support bacterial cell growth and metabolism and that previous work has suggested that soluble protein in media has a significant effect on nanoparticle physicochemical characteristics [46] the CB nanoparticles were characterised in deionised water, TSB (bacterial broth) and in 1% FBS/DMEM (cell culture media) through dynamic light scattering (Table 2). This confirmed that median particle diameter was significantly increased in TSB and 1%FBS/DMEM (531.2 nm in both media at 100 µg/mL CB) compared to water (458.7 nm at 100 µg/mL) at the same concentration. There was also evidence for particle aggregation as confirmed by higher PDI values, suggestive of higher heterogeneity, generated in particles suspended in TSB (0.418 at 10 µg/mL) and 1% FBS/DMEM (0.362 at 10 µg/mL) compared to water (0.246 at 10 µg/mL).

### 3.2. Selection and Optimisation of S. aureus Strain for Co-Exposure

Our previous work confirmed that cellular immune responses to *S. aureus* (ability of neutrophils to kill and phagocytose) was strain dependent [41]. Therefore, to select an appropriate *S. aureus* strain, the same collection (Table 1) was added individually to human lung and skin epithelial cells and the inflammatory chemokine (IL-8) response assessed (Figure 3). Infection of A549 lung and HaCaT skin epithelial cells with eight *S. aureus* strains demonstrated that *S. aureus* SH1000 consistently gave significantly higher inflammatory responses than the other strains tested. This was shown for IL-8 production in HaCaT skin epithelial cells (Figure 3A) and A549 lung epithelial cells (Figure 3B), and IL-6 production in HaCaT skin epithelial cells (Appendix A). These results confirmed the selection of *S. aureus* SH1000 for further use in the current model.

Having selected an *S. aureus* strain with optimal cytokine induction properties, its growth characteristics in 1% FBS/DMEM were confirmed for suitability in co-culture experiments. The growth of *S. aureus* SH1000 in deionised water, TSB, and 1% FBS/DMEM was compared (Appendix A). It was clear that water would not support the growth of *S. aureus*, which showed a delayed logarithmic growth phase. In contrast, growth in 1% FBS/DMEM showed a characteristic logarithmic growth phase (between 2–6 h post inoculation) and a stationary phase (at greater than 9 h post inoculation). Growth in 1% FBS/DMEM showed similar growth characteristics to that in staphylococcal bacterial broth, TSB. Since water did not sufficiently support the growth of *S. aureus*, it was therefore not used for further study.

### 3.3. Effect of CB on S. aureus Growth

The effect of CB (0–100 µg/mL) on *S. aureus* SH1000 growth was investigated in 1% FBS/DMEM and in the staphylococcal growth media, TSB (Figure 4). Preliminary experiments confirmed that concentrations of CB above 10 µg/mL had significant absorbance (OD_600_) that interfered with *S. aureus* growth assays (Appendix A) and thus colony growth assays were used (Figure 4). In 1% FBS/DMEM, CB (0–100 µg/mL) caused a dose-dependent decrease in the growth of *S. aureus* SH1000 as measured by colony forming units (left hand side, Figure 4). This reached significance at 25 µg/mL (compared to untreated control). In contrast, CB had very little effect on the growth of *S. aureus* SH1000 when grown in TSB; no significant differences were observed compared to untreated control (right hand side, Figure 4). Furthermore, CB at 6–100 µg/mL significantly reduced growth in 1%FBS/DMEM compared to equivalent does in TSB. Growth of *S. aureus* in 1% FBS/DMEM and TSB was equal.

This important observation showing that CB (greater or equal to 25 µg/mL) reduced growth of *S. aureus* in 1% FBS DMEM, was investigated by SEM to confirm if cellular toxicity could be observed (Figure 5). SEM imaging clearly confirmed that CB alone (Figure 5A) had a wide distribution of sizes (consistent with the DLS size measurements) and was subject to aggregation. Imaging of *S. aureus* SH1000 showed a typical ‘cocci’ (spherical) appearance as expected (Figure 5B). SEM imaging of CB combined with *S. aureus* SH1000 (Figure 5C), showed areas rich in binding between CB and bacteria. However, in those areas, no evidence of toxicity, such as arrest of binary fission (two cells adhered with septum in place), surface crenulation (breakdown of cell membrane), or decreases in cell size (dormancy) or cell lysis were seen. Indeed, the morphology of *S. aureus* SH1000 in areas rich in CB binding were no different to areas devoid of CB particles. Thus, the decreased growth was not due to toxicity.

### 3.4. Effect of CB and S. aureus SH1000 on (Pro-)Inflammatory Responses in Epithelial Cells

To assess the (pro-)inflammatory response of human epithelial cells (lung and skin) to particle and bacteria, HaCaT and A549 epithelial cells were exposed to CB (0–100 µg/mL) and *S. aureus* SH1000, alone and in combination for 5 and 24 h incubation periods (Figure 6). These exposure periods were consistent with the logarithmic and stationary phases of the *S. aureus* SH1000 growth (Appendix A). Firstly, it was clear that CB alone (white bars) did not cause an increase in the constitutive production of IL-8 (left white bar) in HaCaT (Figure 6A,B) or A549 (Figure 6C,D) epithelial cells at 5 (Figure 6A,C) or 24 h (Figure 6B,D). Although constitutive IL-8 production increased between 5 and 24 h (Figure 6A vs. Figure 6B,C vs. Figure 6D), this did not reach significance. In HaCaT skin epithelial cells at 5 h, *S. aureus* SH1000 alone (CB at 0 µg/mL) significantly increased IL-8 production (Figure 6A). In addition, the *S. aureus* SH1000-induced IL-8 was dose dependently inhibited by the presence of CB, reaching significance at 100 µg/mL (Figure 6A). At 24 h, significantly more IL-8 was produced in the presence of *S. aureus* SH1000 alone compared to 5 h, but also with respect to the uninfected control at 24 h. However, at 24 h, there was only partial inhibition of IL-8 by CB at 25 µg/mL and this effect did not reach significance (Figure 6B).

In comparison, A549 lung epithelial cells (Figure 6C,D) showed a significant increase in IL-8 production in response to *S. aureus* SH1000 at 5 h, compared to uninfected control (Figure 6C, left white bar). By 24 h, IL-8 was significantly enhanced in response to *S. aureus* SH1000 (Figure 6D). Indeed, at 5 and 24 h, *S. aureus* SH1000-induced IL-8 production in lung epithelial cells (Figure 6C,D, black bars) was unaffected by CB over the concentration range studied with consistent significant increases compared to uninfected control at each dose of CB (Figure 6C,D). In the same supernatants, IL-6 and IL-10 were also measured by ELISA. IL-6 production was low, and no differences were detected between treatments (Appendix A) although *S. aureus* SH1000 induced IL-6 at 24 h independently of CB dose (Appendix A). IL-10 was undetectable in all cases.

The production of the small antimicrobial peptide, human β2-defensin (hβD-2), was also determined in response to CB and *S. aureus* SH1000 (Figure 7). At 5 h, hβD-2 could not be detected in either HaCaT or A549 epithelial cells. In contrast, at 24 h, hβD-2, could be detected in supernatants from HaCaT (Figure 7A) and A549 (Figure 7B) epithelial cells. In both cell types, CB had no effect on constitutive hβD-2 production (left white bar) when given alone (white bars). When *S. aureus* SH1000 was given alone, there was an increased production of hβD-2, compared to constitutive production, but the difference was not significant. In HaCaT but not A549 epithelial cells, *S. aureus* SH1000 combined with CB (100 µg/mL) caused a significant increase in hβD-2 compared to *S. aureus* SH1000 alone (Figure 7A). This confirms the ability to detect an antimicrobial peptide in this model system.

Finally, cell viability was also confirmed over the CB doses (0–100 µg/mL) alone (Appendix A) and in combination with *S. aureus* (Appendix A) over the exposure times studied (5 and 24 h). In HaCaT epithelial cells (Appendix A), CB alone caused significantly reduced viability at 50 µg/mL following 5 and 24 h exposure. Cell viability did not decrease below 56%. In A549 epithelial cells (Appendix A), CB alone caused significantly reduced viability at the 100 µg/mL dose after 5 and 24 h. Cell viability did not decrease below 77%. Further assessment of viability in response to combination treatments confirmed that *S. aureus* had very little effect on viability in HaCaT (Appendix A) or A549 cells (Appendix A). In addition, *S. aureus* did not decrease viability beyond that shown for CB at 25 and 100 µg/mL.

## 4. Discussion

The original aim of the current work was to establish an in vitro model whereby skin and lung epithelial cells could be co-exposed to a model particle and an infectious agent to enable their biological impact, namely immune responses, to be determined. It was reasoned that understanding cellular responses (i) to a (nano)particle in the presence of an infection and (ii) the response to infection during (nano)particle exposure is a poorly studied area of research and has implications to understanding such responses in immunosuppressed individuals. This goal was achieved by using HaCaT, keratinocytes, and A549 type II lung adenocarcinoma cells as the epithelial cells of choice due to their robust (pro-)inflammatory responses to a variety of stimuli and their resistance to injury [33,38,39,40]. Carbon black (AROSPERSE^®^ 15 thermal black powder, Evonik Degussa GmbH #BT10506621) was the (nano)particle used in the model due to its presence in a variety of everyday items, such as printer ink and rubber tyres. *S. aureus* was used as the infectious agent due to its importance in skin and lung infections, such as atopic dermatitis and pneumonia, respectively. To the authors’ knowledge, this is one of the first demonstrations of assessing immune parameters in a co-exposure system, where responses from eukaryotic (inflammatory mediators) and prokaryotic (growth) cells in co-culture have been determined following (nano)particle exposure.

Studying the effects of CB (nano)particles in stressed cells has been poorly studied to-date, with most research focusing on ‘healthy’ cells prior to a cyto- or genotoxic stimulus [5,51]. This work challenges that current dogma by studying CB (nano)particle exposure during infection with a human skin and lung pathogen, namely *S. aureus*. The major findings suggest that CB under certain circumstances can attenuate infection induced IL-8 and further enhance infection-induced hβD-2. Very few studies have addressed toxicity during infection. Hussey and colleagues showed that exposure to black carbon (<500 nm, Sigma, Gillingham, UK. # 699632) particles can encourage invasion of *S. aureus* from the nasopharynx to the lower airways to establish infection [37]. In addition, nearly 20 years ago, Jakab et al. confirmed that combinations of CB and formaldehyde or acrolein compromised lung host defence by suppressing killing and phagocytosis of *S. aureus* [36,52]. Furthermore, recent studies of CB exposure in mice studied over 30 days confirm increased autophagy after 7 days, which is reversible after 30 days [53,54]. These studies and our own confirm the importance of studying exposure to injured, stressed, or infected cells. Our viability assessments confirm that over half of the cells in culture remain viable during the treatments given in this study.

SEM and TEM imaging confirmed the distribution of particle sizes assessed by DLS. Although these results should be interpreted with caution due to measurement of dried material in the former and a suspension with the latter, there is a good consistency in size estimation between the techniques. What was more striking was the dramatic effect of biological culture media on (nano)particle physicochemical properties, with increases in CB size following incubation in either the bacterial broth (TSB) or the cell culture media 1% FBS/DMEM.

Media composition was important in determining cell function during bacterial growth experiments. CB had no effect on *S. aureus* SH1000 growth when cultured in TSB bacterial broth. In contrast, 1% FBS/DMEM unmasked the inhibitory effects of CB on bacterial growth. The mechanism underlying this effect is unknown, however a simple explanation may be provided through ‘nutrient restriction’ [55,56,57]. With respect to protein content, TSB, contains tryptone, a digest of the protein casein and contains an assortment of peptides, while 1% FBS/DMEM contains much larger serum proteins, like bovine serum albumin (BSA). The former peptides are much more easily utilised for bacterial metabolism, unlike the larger BSA molecules which are more difficult to breakdown. Interestingly, BSA molecules have been shown to be part of protein corona formation in CB [58] and black carbon [59] and therefore may be sequestered away from the bacteria. Furthermore, others have suggested that many components of serum bind to CB, including BSA, transferrin, apolipoprotein A-1 [60] and fibrinogen [61]. Therefore, our approach to model host, particle and pathogen in cell culture media should be considered when using advanced cell culture systems so that the established effects of ‘adsorption artifacts’ are considered [62].

One major achievement in this study was the successful detection of early induced cytokines (IL-8 and IL-6) and antimicrobial peptides (hβD-2). This is consistent with our previous studies using iron oxide nanoparticles in an epithelial/macrophage co-culture model [63]. In the current study, it was clear that early immune responses, defined by IL-8 production, were generated through *S. aureus* SH1000 stimulation, with an underlying modulation by CB which was time dependent. This is probably not that surprising given previous work showing that *S. aureus* stimulated cytokine and antimicrobial peptide responses in skin and lung epithelial cells and from patients [64,65,66,67,68]. The choice of cytokines and growth factors could be expanded in future work, to include the soluble pattern recognition ‘collectin’ molecules, metalloproteinases, chemokines such as CCL2 and regenerative growth factors such as TGFβ. In these models multiplex assays may provide a better readout of host responses.

It has not escaped our attention that the CB dose of 100 µg/mL when combined with *S. aureus* reduced IL-8 and increased hβD-2 production (compared to *S. aureus* alone) from HaCaT epithelial cells. While this result should be interpreted with caution as CB has been shown to interfere with some ELISA systems [69,70] but not in ours (Appendix A) this result is fascinating and may be the first time this differential effect has been observed with this particle and pathogen combination. Interestingly, differential production of IL-8/hβD-2 seems to be an evolving paradigm in barrier immunity as numerous authors report a similar effect but in anatomically distinct epithelial cell systems using different ‘stressful’ stimuli. Thus, *Proprionibacterium acnes* in skin keratinocytes [71], cigarette smoke in gingival epithelial cells [72] and *Streptococcus pneumoniae* in A549 cells [73] have all demonstrated IL-8/hβD-2 immunomodulation. However, this effect seems even more important in the digestive tract where Probiotics (*Lactobacillus rhamnosus*/*Bifidobacterium longum*) and *Pseudomonas aeruginosa* [74], 1,25-dihydroxyvitamin D3 or statins and *Salmonella typhimurium* [75,76], and heat-killed probiotics (*L. casei*/*L. fermentum*) [77] have all shown similar mechanisms. Exposure at immune barriers is clearly important for determining successful immunity. The cellular signalling pathways underlying these effects are a clear target for future investigation.

The detailed cellular mechanism underlying the immunomodulatory action of CB remains open to interpretation. This is especially difficult to define given the number of potential interactions. Firstly, the role of serum proteins is important for final activity; this includes the proteins present but also their concentration. Indeed, differences in IL-8 output to CB have been observed if the concentration of serum is increased to 10% (compared to the 1% as used here). Secondly, SEM, TEM and DLS suggest that aggregation may also play a role in determining response. Indeed, previous studies in monocytes confirm CB particle size-dependent cytotoxicity [78]. Furthermore, it would be interesting to speculate whether the biphasic profile of IL-8 after 24 h in HaCaT cells was due to particle size and/or aggregation. Thirdly, determination of the cellular targets of CB is vital. Indeed, Vuong and co-workers provide compelling data confirming the particle specific effects of CB in proteomic responses associated with cell death and proliferation pathways in A549 cells [79]. There is also evidence that CB has the potential to bind certain cytokines to influence biological activity [80] or interfere with ELISA technologies [69,70].

Although this study is an advance in host, pathogen, and particle model systems there are also limitations. Firstly, the full mechanisms of CB based immunomodulation was beyond the scope of this work. Secondly, a more equal balance of inflammatory response between pathogen and particle may help further address the mechanism of CB cellular interaction. Finally, our study did not allow further estimation of the safe exposure limits for CB. However, they have been defined on the skin, when used as a colourant, of 20 nm minimum size, and a concentration up to 10% [81]. Limits for lung exposure have been discussed recently [8] and suggest a no-observational-exposure-limit (NOEL) of 1 mg/m^3^ and a LOEL of 7–50 mg/m^3^ depending on the surface area. These values are consistent with exposure in carbon black factory workers in UK and German studies [7]. This converts to doses of 1–50 ng/mL and therefore the doses in this study are relatively high. However, doses are hard to extrapolate to humans because of the significantly greater surface area of the skin and lungs.

The move to study co-exposures is challenging but much more relevant to human pathology and small adaptions in the current model could apply to numerous other particulate and infectious diseases. These include but are not limited to co-infection with influenza and *S. aureus* in the lung [82,83] where a model has recently been considered [84]; exposure of cigarette smoke (passive smoking) and respiratory syncytial virus (RSV) in the lung [85]; human bacterial and viral coinfections with respiratory syncytial viruses [86], not to mention exposures to particles such as titanium, sulphur dioxide, and ozone.

In conclusion, this work confirms the development of a pathogen and particle co-exposure model in human lung and skin epithelial cells. Critically, the work confirms that toxicological testing in healthy versus stressed cells give significantly different responses (Figure 8). This suggests that cell stresses (including infection studied here) should be included in current toxicological testing models to better represent the diversity of cell viabilities found in lung and skin within a general population. Studying particle exposure under stressed, pathological, or infected conditions as compared to traditional exposure in ‘healthy’ cells is a vital addition to understanding the role that particulate exposure has within diseased or immunocompromised patient groups.

## Figures and Tables

**Figure 1 biomedicines-12-00128-f001:**
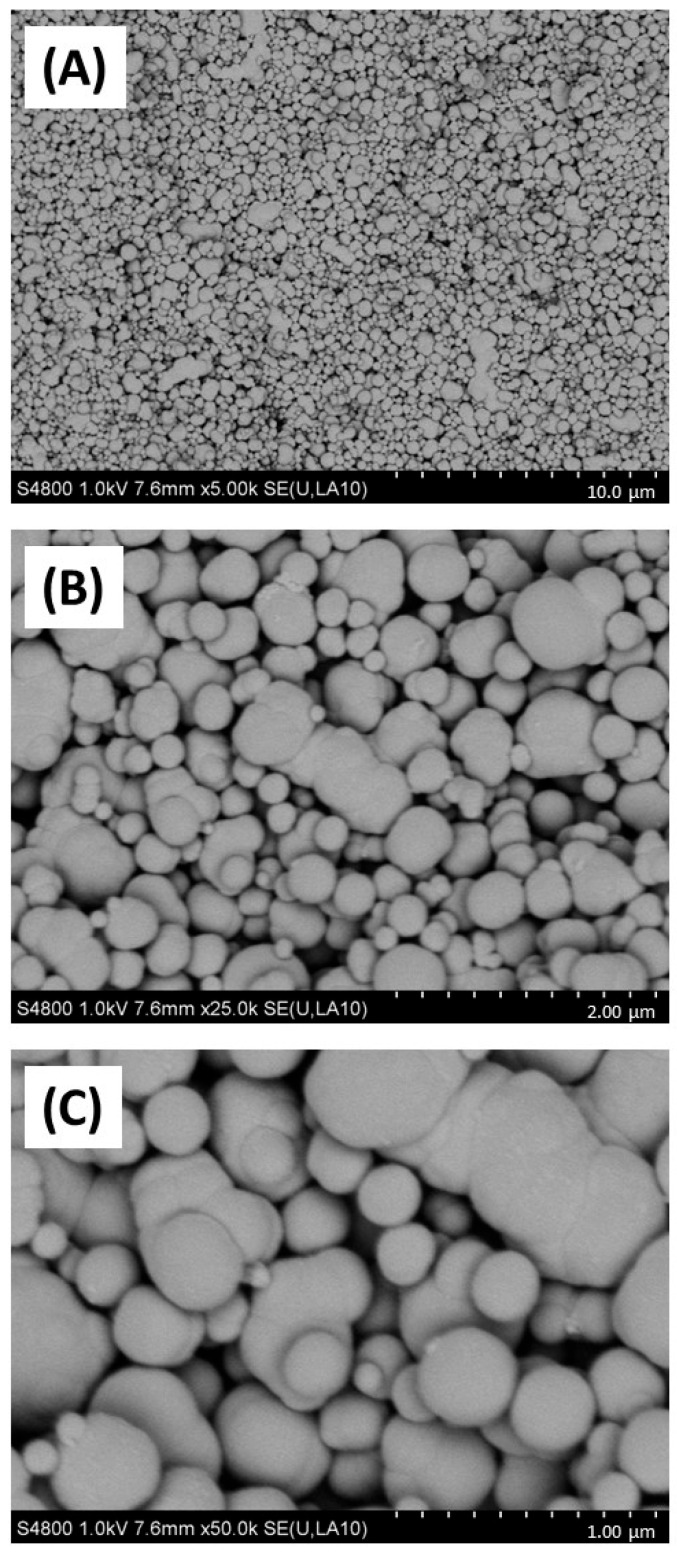
SEM imaging of carbon black. Carbon black (AROSPERSE^®^ 15) was imaged with a Hitachi S4800 high-resolution scanning electron microscope at increasing resolution (**A**–**C**). Particles appeared spherical in shape with clear diversity in particle size. Size scale shown in bottom right of each panel (10 µm, 2 µm and 1 µm in (**A**–**C**), respectively). Images were generated with a Hitachi S4800 microscope, and the original magnifications were (**A**) ×5000, (**B**) ×25,000 and (**C**) ×50,000.

**Figure 2 biomedicines-12-00128-f002:**
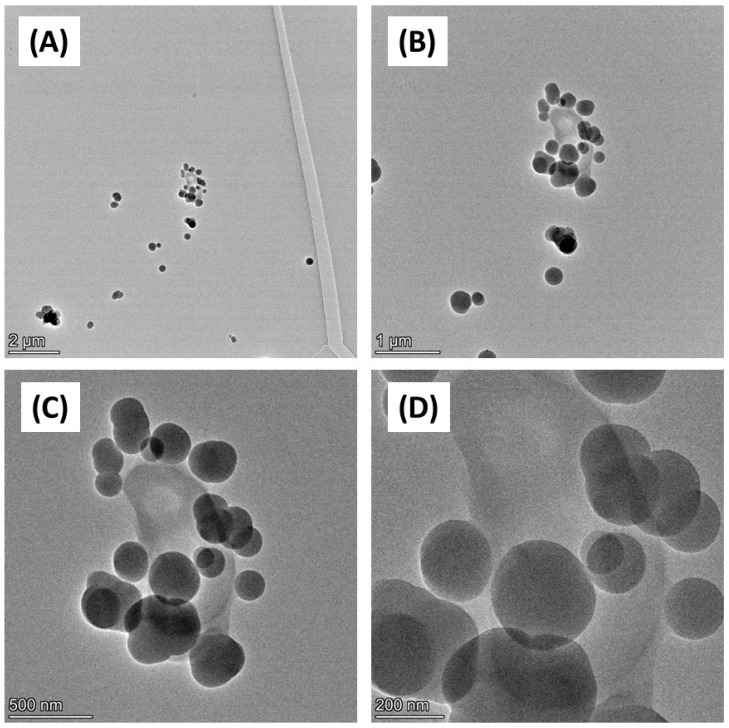
TEM imaging of carbon black. Carbon black (AROSPERSE^®^ 15) was imaged with a FEI Talos F200x G2 Transmission Electron Microscope at increasing resolution (**A**–**D**). Scale bar on the bottom left of each panel (2 µm, 1 µm, 500 nm and 200 nm in (**A**–**D**), respectively). Images were generated with a FEI Talos F200x G2 microscope, and the original magnifications were (**A**) ×4300 (**B**) ×11,000 (**C**) ×28,500; and (**D**) ×58,000.

**Figure 3 biomedicines-12-00128-f003:**
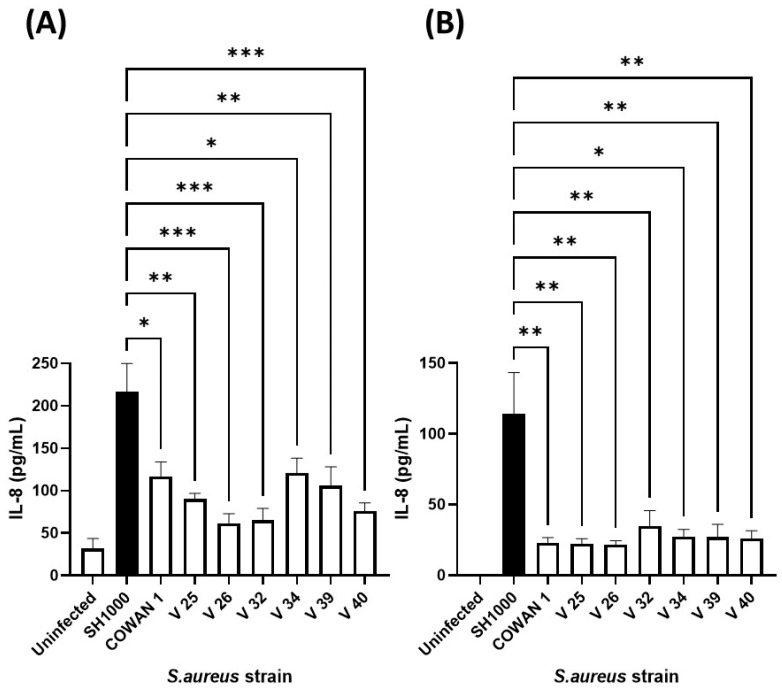
Selection of IL-8-inducing *S. aureus* strain. Human HaCaT skin (**A**) and A549 lung (**B**) epithelial cells were stimulated with eight strains of *S. aureus* for six hours. Supernatants were collected and the concentration of IL-8 determined by ELISA. Results are expressed as the mean ± SEM of 4 experiments (*n* = 4). Differences between treatments were calculated using an ANOVA multiple comparison test with a Tukey’s post hoc test with * *p* < 0.05, ** *p* < 0.01, *** *p* < 0.001, levels considered significantly different. Black bar shows *S. aureus* SH1000 response.

**Figure 4 biomedicines-12-00128-f004:**
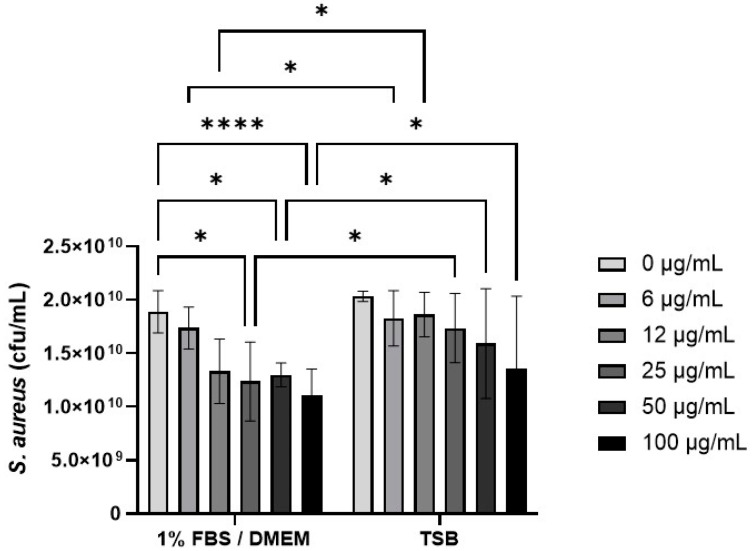
The effect of carbon black on the growth of *S. aureus* SH1000. Standardised suspensions of *S. aureus* and carbon black (AROSPERSE^®^ 15) in 1% FBS/DMEM or TSB (0–100 µg/mL) were incubated for 5 h to assess growth. Then, dilutions were plated onto blood agar and incubated for 24 h. Then colony counts were determined. Results are expressed as Mean ± SEM of 6 experiments (*n* = 6). Differences between treatments were calculated using a two-way ANOVA multiple comparison test with a Tukey’s post hoc test. A * *p* < 0.05, **** *p* < 0.0001 were considered significantly different.

**Figure 5 biomedicines-12-00128-f005:**
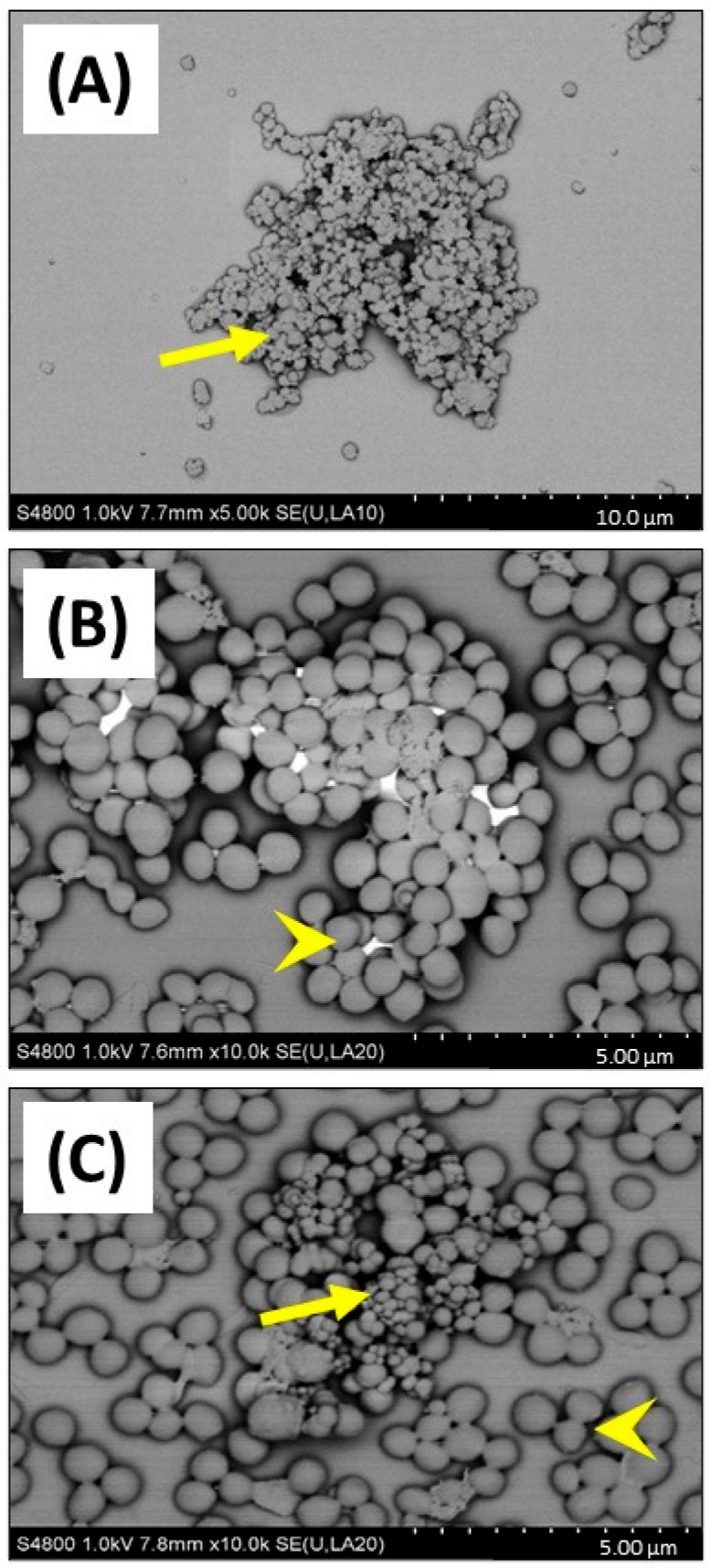
SEM imaging of carbon black and *S. aureus* SH1000. SEM was used to investigate carbon black (AROSPERSE^®^ 15) and *S. aureus* interactions, aggregation, and binding. (**A**) CB (25 µg/mL) alone, (**B**) *S. aureus* SH1000 alone, (**C**) CB (25 µg/mL) and *S. aureus* SH1000. Yellow arrows show carbon black particles. Yellow arrowheads show *S. aureus* SH1000 bacteria. Size scale shown in bottom right of each panel. Images were generated with a Hitachi S4800 microscope with original magnification ×10,000.

**Figure 6 biomedicines-12-00128-f006:**
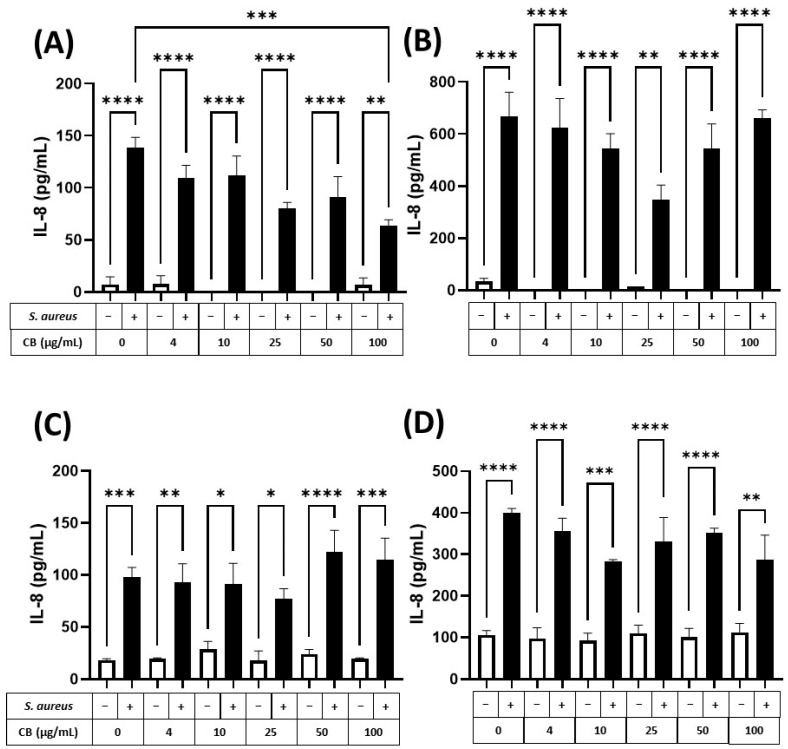
The effect of carbon black and *S. aureus* SH1000 on IL-8 production in epithelial cells. Human HaCaT skin (**A**,**B**) and A549 lung (**C**,**D**) epithelial cells were stimulated with increasing concentrations of carbon black (0, 4, 10, 25, 50, and 100 µg/mL) in combination with *S. aureus* for 5 (**A**,**C**) and 24 (**B**,**D**) hours. Supernatants were collected and the concentration of IL-8 determined by ELISA. Results are expressed as the mean ± SEM of 4 experiments (*n* = 4). Differences between treatments were calculated using an ANOVA multiple comparison test with a Tukey’s post hoc test with * *p* < 0.05, ** *p* < 0.01, *** *p* < 0.001 **** *p* < 0.0001 levels considered significantly different. Black and white bars represent treatments with and without *S. aureus* SH1000, respectively.

**Figure 7 biomedicines-12-00128-f007:**
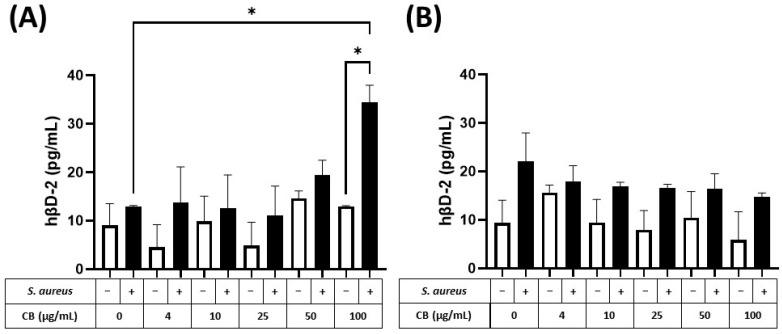
The effect of carbon black and *S. aureus* SH1000 on hβD-2 production in epithelial cells. Human HaCaT skin (**A**) and A549 lung (**B**) epithelial cells were stimulated with increasing concentrations of carbon black (0, 4, 10, 25, 50, and 100 µg/mL) in combination with *S. aureus* for 24 h. Supernatants were collected and the concentration of hβD-2 determined by ELISA. Results are expressed as the mean ± SEM of 4 experiments (*n* = 4). Differences between treatments were calculated using an ANOVA multiple comparison test with a Tukey’s post hoc test with * *p* < 0.05 level considered significantly different. Black and white bars represent treatments with and without *S. aureus* SH1000, respectively.

**Figure 8 biomedicines-12-00128-f008:**
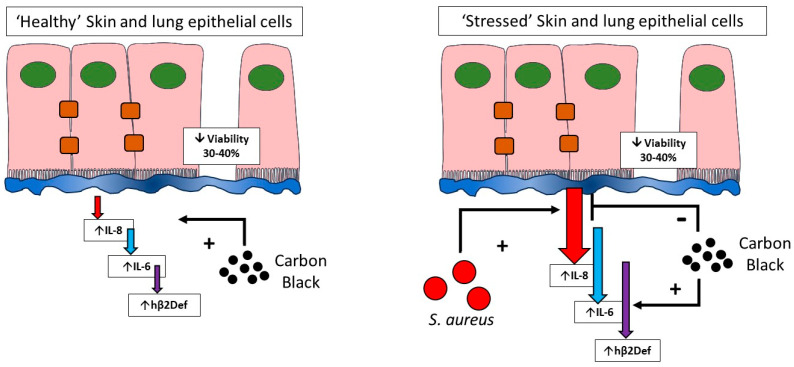
Toxicological testing in healthy and stressed cells. This paper confirms that healthy cells remain relatively resistant to inflammatory cytokine induction by carbon black, but sensitive to cytotoxicity (**left**). In contrast, cells that have undergone infection stress (**right**), where inflammatory output is high unmasks differential effects of carbon black on cytokine output but no further decreases in cell viability. The influence of cell stress should be considered during toxicological testing of nanoparticles.

**Table 1 biomedicines-12-00128-t001:** Strains of *S. aureus* used in this study. Eight *S. aureus* strains were used in this study. They included two reference strains and six strains clinically defined as VAP or non-VAP from a previous study.

*S. aureus* Strain	Source	References
SH1000	Reference/Control	[43,44]
Cowan 1	Reference/Control ATCC 12598/NCTC 8530	[45,46,47,48]
VAP 25	Human BAL	[41,42]
VAP 26	Human BAL	[41,42]
VAP 32	Human BAL	[41,42]
VAP 34	Human BAL	[41,42]
VAP 39	Human BAL	[41,42]
VAP 40	Human BAL	[41,42]

**Table 2 biomedicines-12-00128-t002:** Carbon black physicochemical parameters in media. Carbon black (AROSPERSE^®^ 15) at final concentrations of 4, 10, 25, 50, and 100 µg/mL were suspended in deionised water, 1% FBS DMEM, and TSB, prior to analysis by Zetasizer. Parameters measured include size (nm), median particle population size (nm), hydrodynamic radius (nm), and polydispersity index (PDI).

Carbon Black (μg/mL)	Dispersant/ Solvent	Size Range (nm)	Median Size Population (nm)	Z-Average (nm)	Polydispersal Index
4	Water	220.0–615.0	342	436.8	0.331
10	Water	164.2–955.4	396.1	422.9	0.246
25	Water	190.1–5560.0	487.7	409.9	0.190
50	Water	141.8–5560.0	396.1	405.9	0.175
100	Water	164.2–6439.0	458.7	410.4	0.165
4	TSB	255.0–712.0	342.0	635.3	0.546
10	TSB	255.0–825.0	396.1	625.5	0.418
25	TSB	220.2–955.4	458.7	610.9	0.327
50	TSB	295.3–825.0	458.7	644.5	0.383
100	TSB	295.3–1106.0	531.2	669.6	0.319
4	DMEM with 1% FBS	220.2–825.0	396.1	509.8	0.398
10	DMEM with 1% FBS	190.1–955.4	396.1	530.2	0.362
25	DMEM with 1% FBS	220.2–1106.0	458.7	518.8	0.251
50	DMEM with 1% FBS	220.2–1281.1	531.2	529.9	0.223
100	DMEM with 1% FBS	220.2–1281.1	531.2	511.8	0.163

## Data Availability

The data presented in this study are available on request from the corresponding author.

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
