# Peer review of "An In Vitro Model to Assess Early Immune Markers Following Co-Exposure of Epithelial Cells to Carbon Black (Nano)Particles in the Presence of S. aureus: A Role for Stressed Cells in Toxicological Testing"

_biomedicines, 2024, doi:10.3390/biomedicines12010128_

Round 1
Reviewer 1 Report
Comments and Suggestions for Authors
The authors have investigated the impact of CB exposure during S. aureus infection stress in A549 (lung) and HaCaT epithelial cells. The authors have demonstrated that CB significantly inhibited S. aureus growth but CB did not induce cytokines or antimicrobial peptides from lung and skin epithelial cells. Furthermore, S. aureus induced a robust interleukin (IL)-8 response in both lung and skin epithelial cells. The authors have confirmed toxicological testing in healthy versus stressed cells gives significantly different responses. Overall, this work can inspire more material design ideas of CB-based nanomaterials for antimicrobial and anticancer applications. Therefore, I would like to recommend this work to publish in Biomedicines. Below are some comments for the authors.
1. For supplementary Figure S1 (Energy Dispersive X-ray (EDX) spectroscopy of Carbon Black), please provide the correct spectra. The authors have used wrong images.
2. Please examine the supplementary Figures. The supplementary Figures are obvious mistakes.
3. For the introduction “Staphylococcus aureus is a major pathogen in the lungs and skin, causing significant disease burden to health services worldwide”, more references could be cited to broaden the introduction.
https://doi.org/10.1186/s12951-023-02208-3
Author Response
Reviewer 1
The authors have investigated the impact of CB exposure during S. aureus infection stress in A549 (lung) and HaCaT epithelial cells. The authors have demonstrated that CB significantly inhibited S. aureus growth but CB did not induce cytokines or antimicrobial peptides from lung and skin epithelial cells. Furthermore, S. aureus induced a robust interleukin (IL)-8 response in both lung and skin epithelial cells. The authors have confirmed toxicological testing in healthy versus stressed cells gives significantly different responses. Overall, this work can inspire more material design ideas of CB-based nanomaterials for antimicrobial and anticancer applications. Therefore, I would like to recommend this work to publish in Biomedicines. Below are some comments for the authors.
- For supplementary Figure S1 (Energy Dispersive X-ray (EDX) spectroscopy of Carbon Black), please provide the correct spectra. The authors have used wrong images.
Renumbered SUPP Figures
- Please examine the supplementary Figures. The supplementary Figures are obvious mistakes.
Checked supplementary FIGS and Citations
- For the introduction “Staphylococcus aureus is a major pathogen in the lungs and skin, causing significant disease burden to health services worldwide”, more references could be cited to broaden the introduction.
Added two more refs on immune evasion (new refs 23 and 24).
Reviewer 2 Report
Comments and Suggestions for Authors
Dear Authors
Your manuscript is of great value in the field, as there is indeed a problem of considering symbionts when assessing any impacts. Your test system perfectly shows that the presence of microbiota should not be neglected; it can radically change the outcome of events.
I have a few comments for your work, some of them quite significant, in my opinion. I’m sure you can correct them to make the article more accessible to readers
1) Line 51 – are there any other common cancerogens from this group? Give 2-3 of the most common ones to form a reference frame for the reader
2) line 67 - a few words about the role of the LL-37 protein
3) line 81-89 – the mechanism is well studied, so it would be cool to draw a diagram/drawing as a bonus to the verbal description
4) line 94 – what exactly effects did Hussey and co see? positive or negative? indicate the main result you are referring to, otherwise you will have to turn to the original source
5) make all tables in the form of text tables, not figures. And remove the inscription from it, it’s already in the description above. so far the quality of the drawing is very low and unreadable
6) line 127 – what motivates the choice of particle concentration range? What doses does the average person deal with? What about a person at risk?
7) line 145 onwards – what motivated the choice of time points 5 and 24 hours?
8) Section 2.5 - what motivated the choice of cell lines? What limitations do these model lines have?
9) line 183 - streptomycin concentration in µg/ml
10) all photographs from TEM, SEM (1, 2, 5) are of very low quality. Remove the captions with the number in the picture, you already have this in the description. All scales should be made larger and the font should be bold. In general, it is ideal to make the sc figures now
11) figures 3, 4, 6, 7 – remove the signature from the figure itself, that this “figure …” is also visible in the text description. Very small font, nothing is visible or clear
12) in the discussion section, make a figure/diagram with the most important conclusion from your research
Author Response
Reviewer 2
Dear Authors
Your manuscript is of great value in the field, as there is indeed a problem of considering symbionts when assessing any impacts. Your test system perfectly shows that the presence of microbiota should not be neglected; it can radically change the outcome of events.
I have a few comments for your work, some of them quite significant, in my opinion. I’m sure you can correct them to make the article more accessible to readers
1) Line 51 – are there any other common cancerogens from this group? Give 2-3 of the most common ones to form a reference frame for the reader
Included titanium dioxide and nickel (now line 55)
2) line 67 - a few words about the role of the LL-37 protein
Added sentences on the killing effect on S. aureus, E. coli and rhinovirus (now line 72).
3) line 81-89 – the mechanism is well studied, so it would be cool to draw a diagram/drawing as a bonus to the verbal description
We have incorporated S. aureus into the final figure (point 12 below) rather than create one specifically for S. aureus here which would not include nanoparticles. We think the conclusion figure is much more relevant.
4) line 94 – what exactly effects did Hussey and co see? positive or negative? indicate the main result you are referring to, otherwise you will have to turn to the original source
Added detail on increasing thickness of biofilms, doses of antibiotics and the invasiveness of lung infections (line 100/1).
5) make all tables in the form of text tables, not figures. And remove the inscription from it, it’s already in the description above. so far the quality of the drawing is very low and unreadable
Tables updated and edited. However, they have been added as pictures because they tend to move and become unformatted.
6) line 127 – what motivates the choice of particle concentration range? What doses does the average person deal with? What about a person at risk?
This information was described in the discussion (lines 560-564) under limitations.
7) line 145 onwards – what motivated the choice of time points 5 and 24 hours?
The time points are related to the ‘logarithmic’ and ‘stationary’ phases of S. aureus bacteria (explained in lines 168/9).
8) Section 2.5 - what motivated the choice of cell lines? What limitations do these model lines have?
Carbon black’s major routes for exposure are the skin and lung. We therefore used epithelial cells from the skin and lung. Specifically, HaCaT and A549 cells lines were chosen because we understand their induction kinetics with respect to S. aureus and being cell lines they were likely to give relatively consistent responses, which is essential for toxicological testing. In discussion (lines 453-454)
9) line 183 - streptomycin concentration in µg/ml
Streptomycin concentration is noted in U/ml but we have changed to ug/ml. In addition, the penicillin is actually in U/ml so we have also added the Fisher cat number (15140122, line 212)
10) all photographs from TEM, SEM (1, 2, 5) are of very low quality. Remove the captions with the number in the picture, you already have this in the description. All scales should be made larger and the font should be bold. In general, it is ideal to make the sc figures now
Figures increased and captions removed.
11) figures 3, 4, 6, 7 – remove the signature from the figure itself, that this “figure …” is also visible in the text description. Very small font, nothing is visible or clear
The title words ‘Figure 3, 4, 6 and 7 have been removed and the figures increased in size.
12) in the discussion section, make a figure/diagram with the most important conclusion from your research
Extra figure designed and added to the document to summarise the major findings (Figure 8)
Reviewer 3 Report
Comments and Suggestions for Authors
This paper deals with the establishment of an interesting in vitro cell model comprising lung or skin epithelial cells, carbon black and Staphylococcus aureus to simulate everyday situations of exposure to potential anthropogenic contaminants and/or infections.
The study is well designed, comprises adequately selected comprehensive laboratory techniques and provides a large set of data which have the potential to serve as the basis for future studies.
I have only a handful comments that could be taken into consideration:
- I liked the rationale for each section of the experimental design that was provided in the Discussion section (such as the cell type and bacterium used, etc.). Perhaps, several thoughts could be moved from the Discussion to the Introduction to strengthen the aims and objectives stated.
- I was also wondering, why IL-6, IL-8 and IL-10 were selected for the ELISA assays? As stated in the article, the immune response is a complex reaction that comprises a rather large set of molecules involved. The first cytokines that would come to my mind are perhaps IL-1, TNF alpha and IFN gamma. Maybe, multiplex assays could be performed to study more cytokines that could also be involved in the stimulation of inflammation, and thus hint any mechanisms of action carbon black and S. aureus may come along with a co-exposure (such as epidermal Growth Factor, Monocyte Chemotactic Protein-1 or Vascular Endothelial Growth Factor).
- What would be a possible explanation for the observation that only IL-8 was seen to increase following the co-exposure whilst changes in IL-6 were recorded only in the case of S. aureus infection, and IL-10 did not respond?
- What was the rationale for selecting beta defensin as the representative of the antimicrobial proteins? Would there be any endogenous proteins with antibacterial properties that would deserve attention in future studies?
- Another suggestion to “measure” the rate of damage caused by S. aureus in the presence or absence of carbon black could be molecules that are released by the bacterium, and which are regarded as factors of virulence, such as enterotoxins, lipoteichoic acid, and toxic shock syndrome toxin 1.
On a minor note, please revise the list of References according to the requirements of the Journal, and please increase the figures within the manuscript.
Author Response
Reviewer 3
This paper deals with the establishment of an interesting in vitro cell model comprising lung or skin epithelial cells, carbon black and Staphylococcus aureus to simulate everyday situations of exposure to potential anthropogenic contaminants and/or infections.
The study is well designed, comprises adequately selected comprehensive laboratory techniques and provides a large set of data which have the potential to serve as the basis for future studies.
I have only a handful comments that could be taken into consideration:
- I liked the rationale for each section of the experimental design that was provided in the Discussion section (such as the cell type and bacterium used, etc.). Perhaps, several thoughts could be moved from the Discussion to the Introduction to strengthen the aims and objectives stated.
In lines 158-162 the paragraph has been adjusted to improve the aims and objectives within the introduction.
- I was also wondering, why IL-6, IL-8 and IL-10 were selected for the ELISA assays? As stated in the article, the immune response is a complex reaction that comprises a rather large set of molecules involved. The first cytokines that would come to my mind are perhaps IL-1, TNF alpha and IFN gamma. Maybe, multiplex assays could be performed to study more cytokines that could also be involved in the stimulation of inflammation, and thus hint any mechanisms of action carbon black and S. aureus may come along with a co-exposure (such as epidermal Growth Factor, Monocyte Chemotactic Protein-1 or Vascular Endothelial Growth Factor).
Our previous studies have identified IL-8 (and in some cases IL-6) as a strong and reproducible early responder to nanoparticle stimulation of epithelial cells:
Evans SJ (2019) In vitro detection of in vitro secondary mechanisms of genotoxicity induced by engineered nnoprticles particle fibre toxicology 18 Article number 8 PMID 30760282
Burgum MJ (2021) In vitro primary-indirect genotoxicity in bronchial epithelial cells promoted by industrially relevant few-layer graphene. Small 17(15) e2002551 PMID 32734718
Meldrum, K (2022) Dynamic fluid flow exacerbates the (Pro-)Inflammatory effects of aerosolised engineered nanomaterials in vitro. Nanomaterials. 12(19) Article 3431
In this study we extended our studies to anti-inflammatory IL-10 and the antimicrobial peptide beta 2 defensin which has been reported to be produced by skin and lung epithelial cells (Refs 70-72).
With respect to, IL-1, TNF and IFN. They are really monocyte and lymphocyte derived molecules. Therefore, we might stimulate the epithelial cells with these, but we would be unlikely to measure these.
We have added the reviewers idea on using multiplex assays to further screen for inflammatory biomarkers in the system (lines 670)
- What would be a possible explanation for the observation that only IL-8 was seen to increase following the co-exposure whilst changes in IL-6 were recorded only in the case of S. aureus infection, and IL-10 did not respond?
This is very likely associated with kinetics. IL-8 is induced very early at 5 hours and very robust levels by 24 hours. The time course for IL-6 is much slower in this system (Supp Figure S5). In fact, others confirm that IL-6 and IL-10 is low in A549s (PMID 19825995 and PMID 26987337). In HaCaT cells the pattern is similar but the cells are much more sensitive to calcium which increases cytokine output (PMID 34086734 and PMID 20074471))
- What was the rationale for selecting beta defensin as the representative of the antimicrobial proteins? Would there be any endogenous proteins with antibacterial properties that would deserve attention in future studies?
Three major reasons for this:
- the references within confirming wide expression in different epithelial cells (refs 71-73)
- In addition, there is very good evidence that A549 (ref 73) and HaCaT (ref 71) can produce beta 2 defensin.
- We also had optimised the beta 2 defensin ELISA in skin cells previously
- Another suggestion to “measure” the rate of damage caused by S. aureus in the presence or absence of carbon black could be molecules that are released by the bacterium, and which are regarded as factors of virulence, such as enterotoxins, lipoteichoic acid, and toxic shock syndrome toxin 1.
Actually, this is a very good idea. We were going to look at different clones of S. aureus such as clonal complex 8 or 9 (including USA300). Then also include the toxin work as you suggest, however these strains need an extra level of biosecurity which is not available at our institute at this current time.
On a minor note, please revise the list of References according to the requirements of the Journal, and please increase the figures within the manuscript.
Refs have square brackets in text and then without in the reference list. The ‘Biomedicines’ reference style was used in Mendeley.
Round 2
Reviewer 1 Report
Comments and Suggestions for Authors
The authors have answered all questions raised by the reviewers. Therefore, I would like to recommend this manuscript to publish as its current form in Biomedicines.
Reviewer 2 Report
Comments and Suggestions for Authors
Dear authors,
I agree with all your edits and arguments, I have no more comments